# Enhanced excitability but mature action potential waveforms at mossy fiber terminals of young, adult-born hippocampal neurons in mice

Nicholas P. Vyleta[1] & Jason S. Snyder [1✉]

Adult-born granule neurons pass through immature critical periods where they display enhanced somatic excitability and afferent plasticity, which is believed to endow them with unique roles in hippocampal learning and memory. Using patch clamp recordings in mouse hippocampal slices, here we show that young neuron hyper-excitability is also observed at presynaptic mossy fiber terminals onto CA3 pyramidal neurons. However, action potential waveforms mature faster in the bouton than in the soma, suggesting rapid efferent functionality during immature stages.

---

[1] Department of Psychology, Djavad Mowafaghian Center for Brain Health, University of British Columbia, Vancouver, BC, Canada. ✉email: jasonsnyder@psych.ubc.ca

Adult hippocampal neurogenesis produces a pool of new dentate gyrus neurons that have crucial roles in cognition and emotional behavior[1,2]. Cellular properties of new neurons are intimately linked to developmental stage[3] and central to functional theories is the concept of a "critical period", where immature neurons make important contributions to network signaling through a combination of unique structural and physiological properties. For example, it is well-established that immature neurons display differential plasticity at input synapses[4–10], their survival is modulated by experience[11–16], and they have enhanced somatic excitability[4,17]. A number of theories have therefore proposed that these various forms of developmental plasticity are likely to shape how new neurons are recruited during memory processes and become tuned to external stimuli[18–22].

Despite the wealth of data on the afferent and somatic properties of adult-born neurons, relatively little is known about efferent properties. The dentate gyrus projects to region CA3 via the sparse, but highly powerful, mossy fiber axons. Mossy fiber terminals are among the largest in the brain and they are widely-believed to play a critical role in recruiting CA3 pyramidal neurons during memory encoding[23,24], though they also contribute to mnemonic discrimination during retrieval[25]. Adult-born neurons make functional connections with downstream CA3 pyramidal neurons[26] and there is evidence that mossy fiber axons extend rapidly[27] and display some forms of critical period plasticity[28,29]. However, nothing is known about the electrical properties at the outputs of newborn neurons. These details are important because the shape and dynamics of the presynaptic action potential determines calcium entry and the efficacy of neurotransmitter release[30,31] and therefore could have sizeable effects on information transfer into the CA3 pyramidal cell network. Since action potential signaling at mossy fiber terminals is modulated by passive spread of excitatory postsynaptic potentials from the granule cell body[32], critical period differences in bouton excitability could also impact synaptic efficacy.

To address these questions, here we investigated the development of membrane excitability and action potential properties in adult-born dentate granule neurons and individual mossy fiber terminals at output synapses in the CA3 region of the hippocampus. We used Ascl1$^{CreERT2}$ mice[6,33] of both sexes to birthdate adult-born and neonatally-born dentate granule neurons, and patch-clamp recordings from tdTomato-positive soma and boutons to measure passive and active properties of electrical signaling.

## Results

To verify that, in our hands, young adult-born neurons displayed known critical period properties, we first determined the time-course of maturation of intrinsic excitability. Individual granule cells of specific age (specifically, Days Post-tamoxifen Injection; DPI) were targeted for patch-clamp electrophysiology recordings (Fig. 1a). Consistent with previous studies[6,33], young (4–6w) adult-born granule neurons had increased input resistance, which declined as cells matured to 8+ weeks of age (Fig. 1b–d). In current-clamp experiments, young adult-born neurons fired action potentials with smaller amounts of current injection (Fig. 1e, f). The current threshold for evoking an action potential increased with cell age and converged with that for neonatal-born cells and mature adult-born neurons[34] (Fig. 1f, g). Additionally, the action potential peak amplitude was smaller and the half-duration was longer for young adult-born versus mature adult-born cells (Fig. 1h–k). In contrast to adult-born cells, electrical properties of neonatally-born neurons were constant over the same timeframe (Fig. 1d, g, i, k), indicating that time-dependent differences were not due to changes

in animal age. Collectively, these results are consistent with prior work demonstrating a critical period where immature neurons are more excitable[4,33] and display immature action potential waveforms at the somatic compartment[33,35].

To investigate whether output signaling may be altered by age-dependent changes in excitability, we next performed subcellular patch-clamp recordings from individual mossy fiber terminals of birthdated granule neurons (Fig. 2). Large mossy fiber terminals were found from cells in the full age range examined. Bouton capacitance, which is indicative of bouton size, was estimated from a test pulse in voltage-clamp recordings[30,36] (Fig. 2b) and did not differ between experimental groups (Fig. 2e, f). Input resistance, on the other hand, was increased at terminals from young adult-born granule neurons and was significantly correlated with cell age (Fig. 2c, d). Therefore, the membranes of the boutons from young adult-born neurons were relatively immature, consistent with results at the somatic compartment of those neurons, although the relative reduction in input resistance was less for boutons than for soma (mean $R_{in(8+w)}/R_{in(4-6w)} = 0.66$ vs 0.46, respectively).

We next measured the excitability of the boutons directly by injecting current in current-clamp recordings. All groups of boutons showed the characteristic phenotype of pronounced rectification of voltage changes and firing of only a single action potential (due to the high expression of voltage-activated potassium channels[30,37]) (Fig. 2g). Slopes of the average V–I curves were steeper for boutons from young adult-born neurons, consistent with the elevated input resistance (Fig. S1). Similar to results from somatic recordings (Fig. 1), boutons from young adult-born neurons fired action potentials with less current injection (had lower rheobase), and this minimum current threshold was positively correlated with cell age (Fig. 2h, i). The membrane potential at which boutons fired action potentials (AP threshold), on the other hand, was not different between groups nor did it depend on age of adult-born neurons (Fig. S2). In contrast to the somatic recordings, waveforms of the evoked action potential were similar between groups—neither the amplitude nor the half-duration of action potentials evoked by the largest current injection (70 pA) differed as a function of cell age (Fig. S3). These results confirmed that mossy fiber terminals from young adult-born granule neurons are hyperexcitable in current injection experiments, likely owing to elevated input resistance and therefore greater ability for smaller stimuli to reach AP threshold.

The somatodendritic compartment of a neuron integrates subthreshold afferent inputs to generate an action potential (usually in the proximal axon[38]), and enhanced somatic excitability enables young neurons to fire in response to relatively weak synaptic innervation[39]. In contrast, axonal boutons are activated by orthodromic action potentials[40] and, furthermore, mossy fiber axons have a high density of voltage-activated sodium channels which ensures reliable activation of the boutons[41]. We therefore performed experiments with brief current injections (Fig. 3), which produce action potential waveforms that are similar to those of a traveling action potential[42,43]. Surprisingly, neither the amplitude (Fig. 3b, c) nor the half-duration (Fig. 3d, e), nor the maximal rate of rise[44] (Fig. S4) of action potentials were different between the groups of boutons, nor were they significantly correlated with granule cell age. Out of concern that differences in fast action potentials may be missed due to RC filtering by high series resistance recording electrodes used to record from the boutons (see Methods and Fig. S5), we performed additional experiments at 22 °C (Fig. S6). Action potentials were larger for all groups, consistent with less filtering of the slower signals (although biological factors may also contribute to reduction in AP amplitude with increased temperature[45]) but no differences were observed in waveforms between groups (Fig. S6).

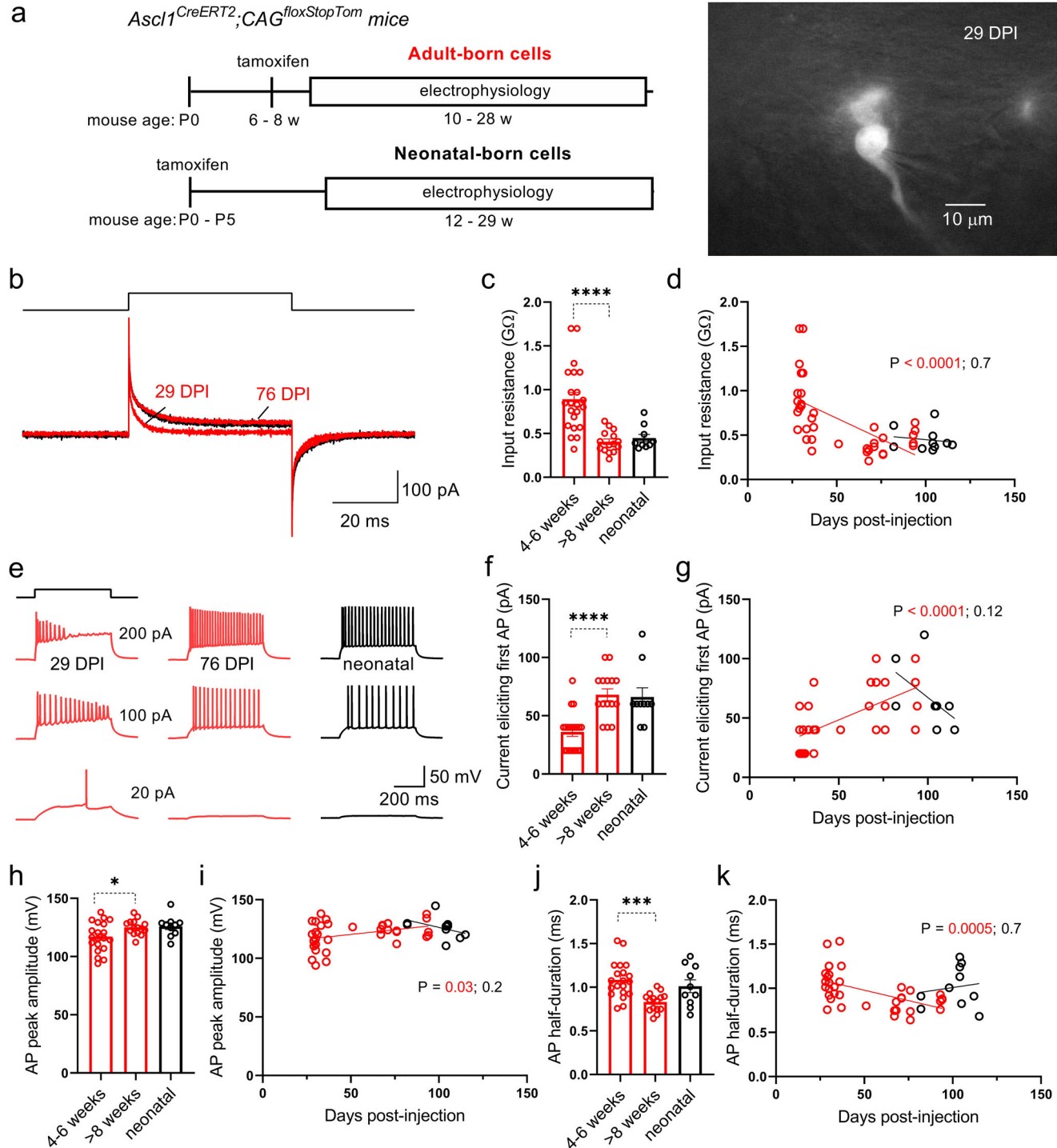

These results indicate that, although membranes of boutons from young adult-born granule neurons were immature (in terms of input resistance), action potentials at those structures had a mature phenotype.

A remarkable property of the action potential at hippocampal mossy fiber terminals is frequency-dependent broadening of the waveform with repetitive signaling, and this is thought to contribute to facilitation of transmitter release during high-frequency trains of action potentials[30]. To further investigate the dynamic capabilities of mossy fiber terminals from adult-born granule neurons at different developmental stages, we evoked high-frequency trains of action potentials in current-clamp recordings (Fig. 4). Action potential broadening at boutons from young

adult-born neurons was pronounced and not different from that at boutons from mature adult- or neonatal-born granule neurons–for either 20- or 100-Hz activation of the boutons. Taken together, these results indicate that the action potential waveform at the outputs of young (otherwise immature) adult-born granule neurons is mature both in terms of shape and dynamics.

## Discussion

Here we report direct evidence for mature action potentials at output terminals from young adult-born granule neurons, in contrast to critical period signaling observed at the somatic

**Fig. 1 Enhanced excitability and immature action potentials in young adult-born granule neurons. a** Experimental timeline for recording from birthdated neurons in Ascl1Cre mice (left) and example fluorescence photomicrograph (right) of a tdTomato-expressing adult-born granule neuron (29 days post-injection) in a whole-cell recording. **b** Response to a test pulse (10 mV, top) in whole-cell recordings from granule neurons of different ages (adult- and neonatal-born in red and black, respectively). Note the reduced steady-state currents in young adult-born neurons, consistent with high input resistance of immature cells. Input resistance values derived from experiments like shown in B for the three different groups of neurons (**c**; one-way ANOVA, $P < 0.0001$, $F_{(2, 44)} = 17$; 4-6w vs 8 + w, ****$P < 0.0001$, Hedges's $g = 1.6$) and plotted as a function of cell age for adult-born granule neurons (**d**). Input resistance declines with cell age for adult-born neurons ($R^2 = 0.37$, $P < 0.0001$, $F_{(1, 36)} = 20$). **e** Example traces from three different neurons showing voltage responses to different amplitudes of current injection (500 ms, top). Note that the young adult-born granule neuron fired an action potential in response to smaller current. **f** Minimum current required to evoke an action potential from experiments like in E for the three groups of neurons. Note that a critical period exists between 4- and 6-weeks of cell age in which neurons are hyperexcitable (one-way ANOVA, $P < 0.0001$, $F_{(2, 42)} = 14$; 4-6w vs 8 + w, ****$P < 0.0001$, Hedges's $g = 1.7$). **g** Minimum current required to evoke an action potential plotted as a function of cell age for adult-born granule neurons. Older neurons require more current to fire action potentials ($R^2 = 0.42$, $P < 0.0001$, $F_{(1, 34)} = 24$). Average peak amplitude (**h**; one-way ANOVA, $P = 0.03$, $F_{(2, 42)} = 3.8$; 4-6w vs 8 + w, *$P = 0.03$, Hedges's $g = 0.73$) and half-duration (**j**; one-way ANOVA, $P = 0.0011$, $F_{(2, 42)} = 8.1$; 4-6w vs 8 + w, ***$P = 0.0003$, Hedges's $g = 1.4$) for action potentials from the three groups, and plotted as a function of cell age for adult-born neurons (**i**, **k**; $R^2 = 0.13$, $P = 0.03$, $F_{(1, 34)} = 5.1$ and $R^2 = 0.30$, $P = 0.0005$). Action potentials in young adult-born neurons have smaller amplitudes and broader durations than in older adult-born neurons. Bars reflect mean ± standard error.

compartment or at input synapses to those neurons. These results have important implications for how the dentate gyrus—containing a diverse population of granule neurons (e.g., young vs old)—processes information and sends electrical signals into the CA3 network. For example, the unique contributions to signaling made by adult-born granule neurons may be confined to the dentate itself, through differential plasticity at entorhinal inputs and integration of those inputs at a hyperexcitable cell body. Rapid development of output signaling may ensure that the information carried by that differential plasticity is faithfully transmitted downstream into the CA3 region. On the other hand, critical period differences in bouton excitability could also be functionally relevant. Subthreshold EPSPs at dentate granule neurons—possibly during theta oscillations[46]—can passively spread to mossy fiber terminals and enhance AP-evoked transmitter release ("combined analog and AP coding"[32]). It is therefore possible that elevated input resistance at young adult-born neurons and their mossy fiber terminals (Figs. 1, 2), combined with a potentially greater axonal length constant (lambda being proportional to the square root of membrane resistance[47]), could increase passive spread of somatodendritic signals. Enhanced analog modulation of transmitter release could therefore increase the computational repertoire of output terminals of young adult-born granule neurons.

How is a mature action potential waveform generated despite relatively immature excitability of the mossy fiber terminal membrane? A high density of voltage-activated sodium and potassium channels[30,41] at mossy fiber terminals might provide a safety factor of sufficient conductance to normalize AP waveforms despite modest differences in passive properties. Future studies measuring sodium and potassium channel densities directly (e.g., with outside-out patch recordings[48]) as a function of granule neuron age may help to clarify this point.

Previously it was shown that excitatory postsynaptic currents (EPSCs) between granule neurons and CA3 pyramidal neurons are normal size by 4 weeks of cell age[29]. However, EPSC size is determined by multiple presynaptic and postsynaptic factors. For example, at the presynaptic terminal an AP of a given waveform must activate calcium channels to produce sufficient calcium influx to activate exocytosis sensors located on the synaptic vesicles that are close to the sites of calcium entry. Several of these factors (AP duration, calcium channel subtype and currents, spatial coupling between calcium channels and synaptic vesicles) have been shown to change during development of a synapse[31,49]. Our study indicates that a key element of presynaptic signaling—the AP waveform—is relatively mature early in the development of synapses from adult-born granule neurons.

In summary, we find that critical period differences exist for some, but not all, aspects of presynaptic physiology. As in the cell body, young neuron terminals are hyper-excitable. However, in contrast to the cell body, young neuron action potentials are mature. It should be noted that our results do not indicate whether other aspects of action potential transmission, such as propagation speed and reliability, are equal at cells of different developmental stages. Furthermore, whether there are developmental changes in other factors that control neurotransmitter release and post-synaptic function should be the subject of future studies, since efficacy of output signaling[50–52] is likely to influence how CA3 pyramidal neurons are recruited during memory encoding and recall.

## Methods

**Animals**. All procedures were approved by the Animal Care Committee at the University of British Columbia and conducted in accordance with the Canadian Council on Animal Care guidelines regarding humane and ethical treatment of animals. Ascl1CreERT2 mice (Ascl1tm1.1(Cre/ERT2)Jejo; JAX 12882v[53]) and Ai14 reporter mice (Gt(ROSA)26Sortm14(CAG-tdTomato)Hze; JAX 7908)[54] were purchased from The Jackson Laboratory, and were crossed to generate offspring that were heterozygous for Ascl1CreERT2 and homozygous for the Cre-dependent tdTomato reporter, as described elsewhere[33] (hereafter, Ascl1CreERT2;tdTomato mice). Mice were maintained on a C57Bl/6J background, housed 5/cage (floor space 82 square inches), with ad lib access to food and water and a 12 h light-dark schedule with lights on at 7 a.m. To induce tdTomato expression in Ascl1+ precursor cells and their progeny, mice were injected intraperitoneally with tamoxifen either neonatally (postnatal day 0–5; ~75 mg/kg, one injection) or during adulthood (6- to 8-weeks-old; 150 mg/kg body weight, one injection/day for up to two days) to permanently label newborn neurons[33]. Adult mice of both sexes were used for electrophysiology experiments between 10- and 29- weeks of age.

**Brain slice preparation**. Mice were anesthetized with sodium pentobarbital (50 mg/kg) before cardiac perfusion with cold cutting solution. Following perfusion, transverse hippocampal slices (350 μm thick) were prepared as described previously[55]. Cutting and storage solution contained (in mM) 87 NaCl, 25 NaHCO₃, 2.5 KCl, 1.25 NaH₂PO₄, 10 glucose, 75 sucrose, 0.5 CaCl₂, 7 MgCl₂, equilibrated with 95% O₂ and 5% CO₂, (pH-adjusted to 7.4 with HCl, ~325 mOsm). Slices were held at 35 °C for at least 60 min before performing experiments at near-physiological temperature (32–34 °C) except for subset at 22 °C as indicated.

**Patch-clamp recordings from tdTom-labeled granule neurons and mossy fiber boutons**. Patch-clamp recordings were made from single granule neurons in the dentate gyrus or mossy fiber terminals in the CA3 region of the hippocampus. Recordings were made in artificial cerebrospinal fluid (ACSF) containing (in mM) 125 NaCl, 25 NaHCO₃, 2.5 KCl, 1.25 NaH₂PO₄, 25 glucose, 1.2 CaCl₂, 1 MgCl₂, equilibrated with 95% O₂ and 5% CO₂, ~320 mOsm. Slices were washed in ACSF for at least 10 min before attempting recordings.

Granule neuron somatic recordings: Recording pipettes were fabricated from 2.0 mm/1.16 mM (OD/ID) borosilicate glass capillaries and had resistance ~5 MΩ with an internal solution containing (in mM): 120 K-gluconate, 15 KCl, 2 MgATP, 10 HEPES, 0.1 EGTA, 0.3 Na2GTP, 7 Na2-phosphocreatine (pH 7.28 with KOH, ~300 mOsm). Current-clamp and voltage-clamp recordings were performed at

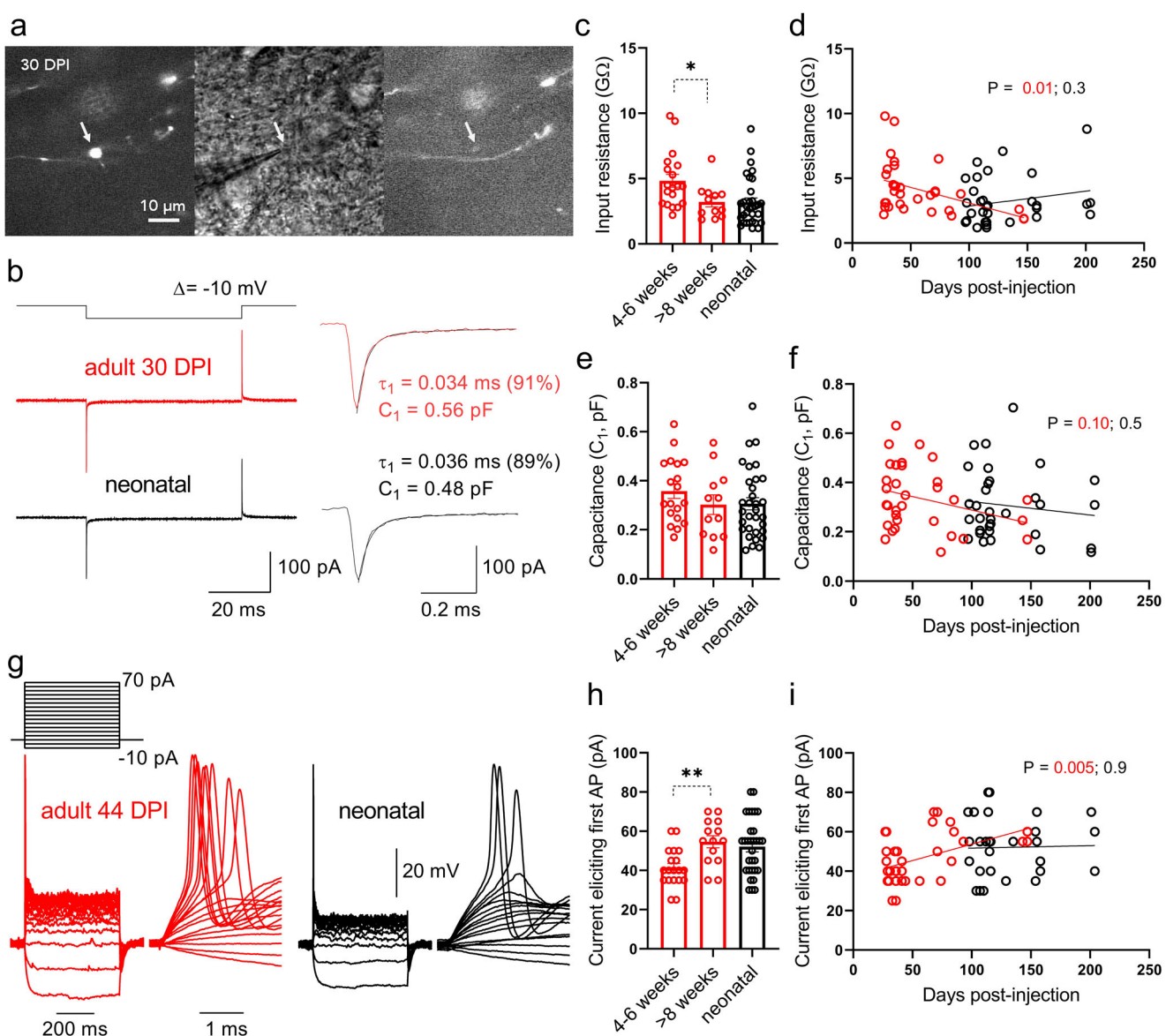

**Fig. 2 Recordings from mossy fiber terminals of birthdated granule neurons reveal a critical period of increased input resistance and enhanced excitability. a** Fluorescence images before (left) and at the end of (right) a targeted patch-clamp recording from a mossy fiber bouton (indicated by arrow) from an adult-born (30 DPI) granule neuron, and corresponding IR-DIC photomicrograph (middle). **b** Current responses to a test pulse (−10 mV, top) evoked in a whole-cell recording from a mossy fiber bouton from either adult- (30 DPI, red traces) or neonatal-born (black traces) granule neuron. Insets: bouton size was estimated by exponential fitting of the capacitive transient currents[30]. Input resistance values determined from the steady-state currents evoked by the test pulse in B for boutons from the three groups of cells (**c**; Kruskal-Wallis, $P = 0.005$; 4–6w vs 8 + w, *$P = 0.04$, Hedges's $g = 0.83$), and plotted as a function of cell age (**d**; $R^2 = 0.18$, $P = 0.015$, $F(1, 30) = 6.7$ for adult-born neurons). Note that input resistance is higher for boutons from young adult-born neurons, consistent with the findings in granule neuron somatic recordings (see Fig. 1). Capacitance values determined from experiments in (**b**) for the three groups of cells (**e**; one-way ANOVA, $P = 0.39$, $F(2, 60) = 0.96$; 4–6w vs 8 + w, $P = 0.28$) and plotted as a function of cell age (**f**; $R^2 = 0.09$, $P = 0.10$, $F(1, 30) = 2.9$ for adult-born neurons). No consistent differences in bouton size were observed between groups. **g** Example voltage responses to pulses of hyperpolarizing or depolarizing current (500-ms, top) for whole-cell recordings from a mossy fiber bouton from either adult- (44 DPI, red) or neonatal-born (black) neuron. Note the characteristic single action potential evoked by suprathreshold pulses. **h** Current required to evoke an action potential from experiments like in g. Note that boutons from 4- to 6-week old adult-born neurons fired with less current, consistent with an enhanced excitability (one-way ANOVA, $P = 0.005$, $F(2, 61) = 5.8$; 4–6w vs 8 + w, **$P = 0.004$, Hedges's $g = 1.2$). **i** Current required to evoke an action potential plotted as a function of cell age for mossy fiber boutons from adult- and neonatal-born granule neurons ($R^2 = 0.22$, $P = 0.005$, $F(1, 32) = 9.0$ for adult-born neurons). Bars reflect mean ± standard error.

−80 mV. Only recordings with high seal resistance (several giga-ohms) and low holding current (less than 50 pA) were included in analyses. For current-clamp recordings, series resistance and pipette capacitance were compensated with the bridge balance and capacitance neutralization circuits of the amplifier, respectively (capacitance neutralization adjusted to ~70% of Cp fast determined in voltage-clamp). A total of 48 granule neuron recordings from 23 animals are reported here.

Mossy fiber terminal recordings: Recording pipettes were fabricated from 2.0 mm/0.7 mm (OD/ID) thick-walled borosilicate glass tubing and had resistance values of ~15–20 MΩ. Pipette solution contained (in mM) 140 KCl, 2 MgATP, 4 NaCl, 10 HEPES, 10 EGTA, pH 7.28, ~310 mOsm. Only recordings with seal resistance >5 GΩ were included in analyses. Whole-cell voltage-clamp recordings were made from identified boutons at −80 mV, and current-clamp recordings made at −70–75 mV. The soluble tdTomato fluorophore provides an excellent

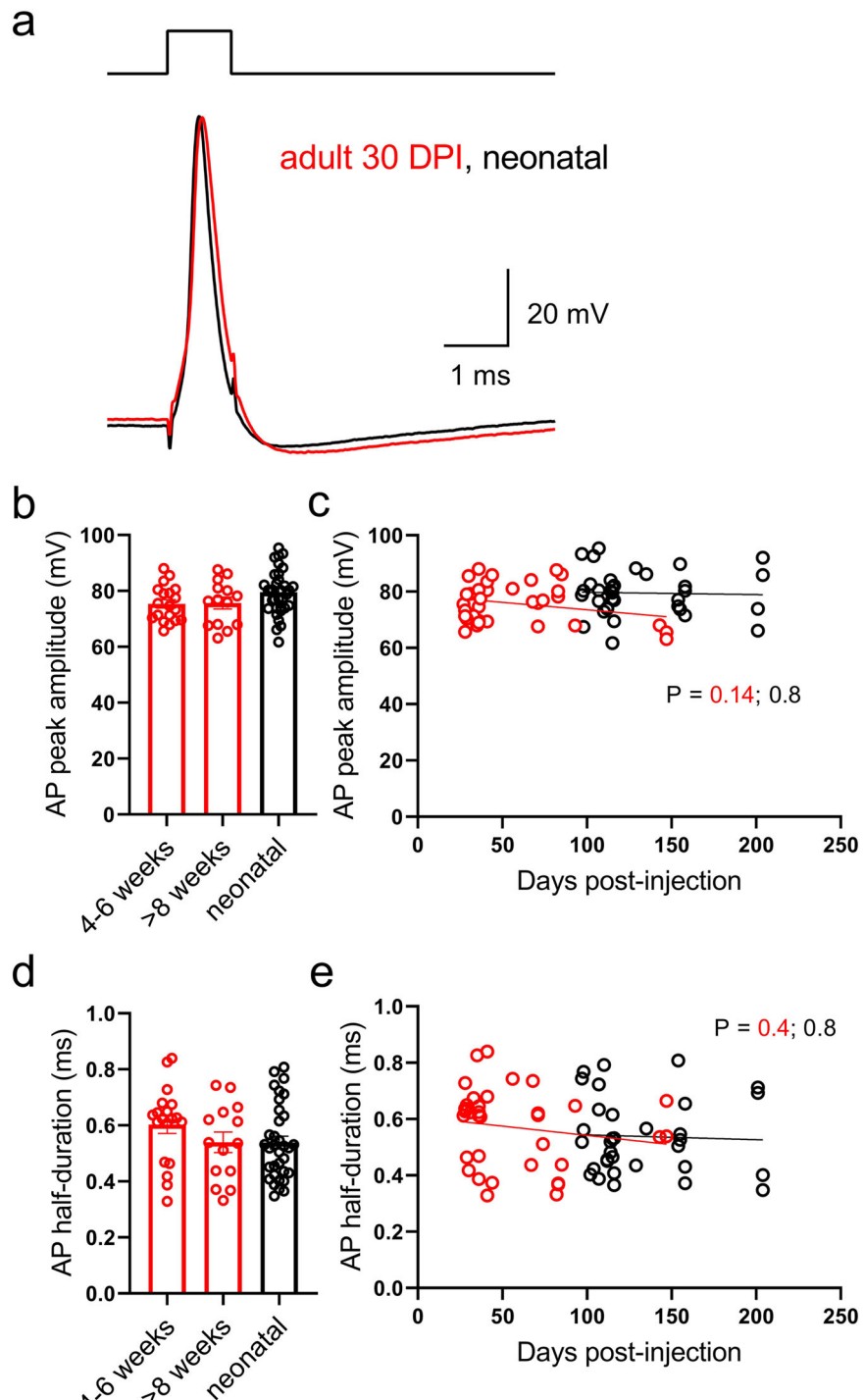

**Fig. 3 Mature action potentials at boutons from young adult-born granule neurons. a** Example action potentials recorded in mossy fiber boutons from either adult- (30 DPI, red) or neonatal-born (black) granule neuron evoked by a brief current injection (200 pA, 1 ms), which approximates orthodromic action potentials that propagate from the granule neuron[42,43]. Average peak amplitude (**b**; one-way ANOVA, $P = 0.10$, $F(2, 64) = 2.4$; 4–6w vs 8 + w, $P = 0.92$) and half-duration (**d**; one-way ANOVA, $P = 0.22$, $F(2, 64) = 1.5$; 4–6w vs 8 + w, $P = 0.19$) of action potentials evoked in boutons from the three groups of cells, and plotted as a function of cell age (**c**, **e**; $R^2 = 0.06$, $P = 0.14$, $F(1, 32) = 2.3$ and $R^2 = 0.03$, $P = 0.36$, $F(1, 32) = 0.87$, respectively for adult-born neurons). Note that no consistent differences were observed between the groups, consistent with mature action potential waveforms at the output boutons of immature adult-born granule neurons. Bars reflect mean ± standard error.

signal for verification of the recorded mossy fiber terminal (to ensure specific targeting)—tdTomato fluorescence disappears over tens of seconds after achieving whole-cell configuration (Fig. 2a), consistent with dialysis of the bouton by the pipette solution[56]. Notably, mossy fiber terminals recorded from here in mice were smaller than previously reported for rats (compare whole-cell capacitance values here of ~0.35 pF to 1.2 pF[30] and 1.7 pF[57], as well as greater input resistance in all subsets of our recordings). In current-clamp recordings, care was taken to

maximally compensate series resistance and pipette capacitance ($C_{pipette}$, which was minimized by use of thick-walled capillaries) with the bridge-balance and capacitance neutralization circuits of the amplifier, respectively, to optimally record action potentials from small structures (capacitance neutralization adjusted to 104% (mean) of $C_{pipette(fast)}$ fast determined in voltage-clamp[36]). Inevitably some residual capacitance and relatively high-resistance recording electrodes leads to filtering of fast signals, especially for small structures like boutons[36,58] and as

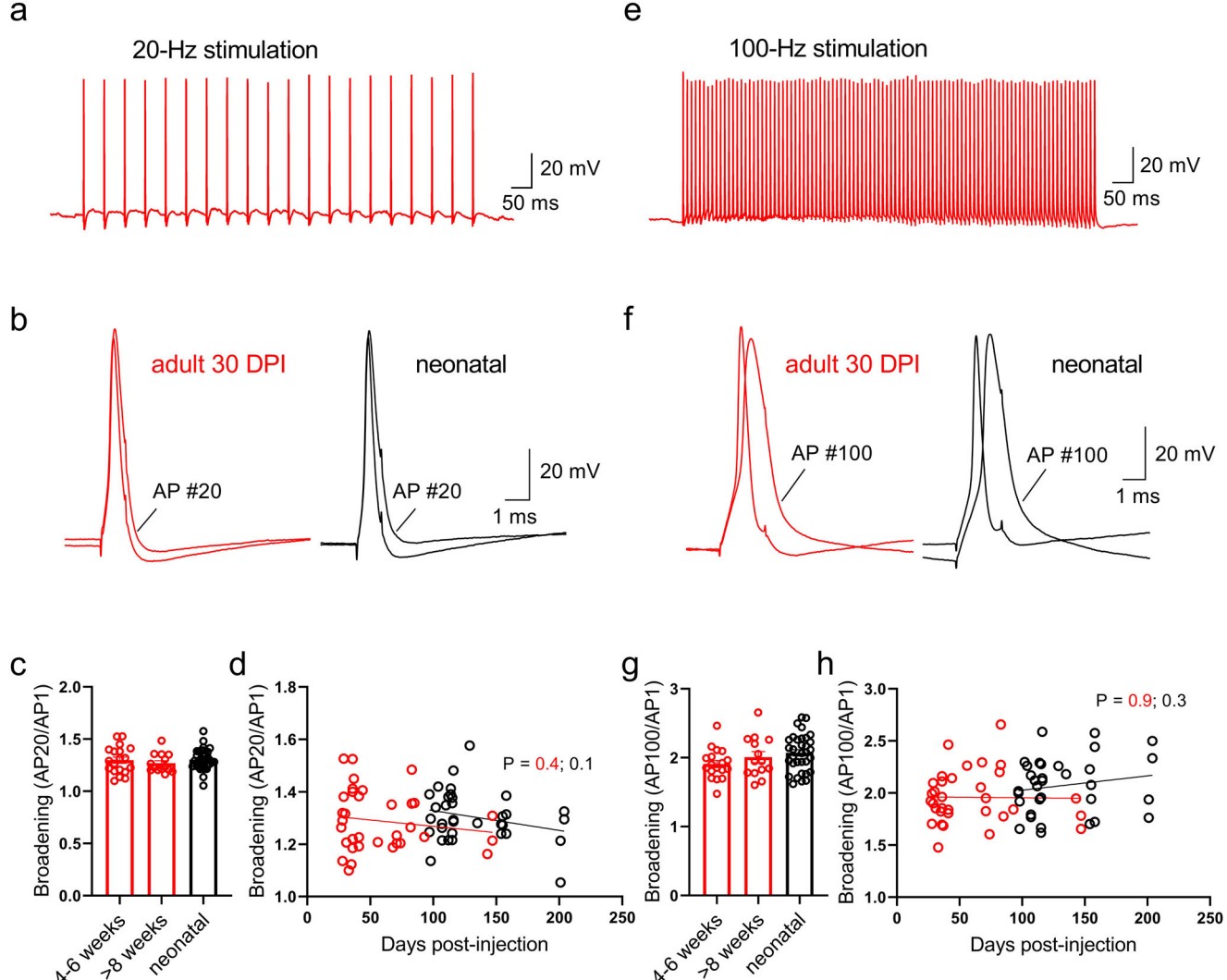

**Fig. 4 Mature activity-dependent broadening of action potentials in boutons from young adult-born granule neurons. a** Trains of action potentials evoked at 20-Hz in a mossy fiber bouton from an adult-born granule neuron (30 DPI; stimulus 200 pA, 1 ms). **b** The first and twentieth action potentials overlaid for bouton recordings from either adult- (30 DPI, red) or neonatal-born (black) granule neuron, showing characteristic broadening. Ratios of half-durations for the twentieth and first action potentials in a 20-Hz train for boutons from the three groups of cells (**c**; one-way ANOVA, $P = 0.56$, $F(2, 64) = 0.58$; 4-6w vs 8 + w, $P = 0.44$) and plotted as a function of cell age (**d**; $R^2 = 0.02$, $P = 0.41$, F(1, 32) = 0.70 for adult-born neurons). **e** Trains of action potentials evoked at 100-Hz in a mossy fiber bouton from an adult-born granule neuron (30 DPI; stimulus 100 pA, 2-ms). **f** The first and one-hundredth action potentials overlaid for bouton recordings from either adult- (30 DPI, red) or neonatal-born (black) granule neuron. Ratios of half-durations for the one-hundredth and first action potentials in a 100-Hz train for boutons from the three groups of cells (**g**; one-way ANOVA, $P = 0.12$, $F(2, 62) = 2.2$; 4-6w vs 8 + w, $P = 0.29$) and plotted as a function of cell age (**h**; $R^2 = 0.0002$, $P = 0.94$, F(1, 31) = 0.007 for adult-born neurons). Note that boutons from all groups of neurons show equivalent broadening of action potential waveforms with high-frequency stimulation. Bars reflect mean ± standard error.

expected, AP peak amplitudes and half-durations were significantly correlated with series resistance of the recordings (Fig. S5) but not correlated with pipette capacitance (due to capacitance neutralization; all mossy fiber recordings pooled). Critically, neither pipette capacitance nor series resistance were significantly different between groups (Fig. S5, mean $C_{pipette} = 6.9$, 6.8, and 6.9 pF; mean $R_s = 94$, 92, and 95 MΩ for 4-6w, 8 + w, and neonatal recordings, respectively). Importantly, the action potential broadening measurements provide an internal control for the temporal resolution of our recording system—our measurements of differences in AP duration are quantitatively equivalent to broadening reported previously[30] (Fig. 4)—demonstrating that our measurements are within the dynamic range of our recording system. A total of 93 mossy fiber terminal recordings from 67 mice are reported here.

**Data acquisition and analysis**. Data were acquired with a Multiclamp 700B amplifier (Axon Instruments) controlled by pClamp 10 software (Molecular Devices), low-pass filtered at 10 kHz and sampled at 100 kHz. Data were analyzed with pClamp, IgorPro (Wavemetrics), and Stimfit[59]. Input resistance was measured

from a test pulse (10 mV for granule cells, −10 mV for mossy fiber terminals) in voltage-clamp. For granule cell recordings, the peak amplitude and half-duration for the first 10 action potentials evoked by current injection steps were averaged for each cell (Fig. 1). For mossy fiber terminal recordings, peak amplitude and half-duration from three separate action potentials evoked by brief current injection (200 pA, 1 ms) were used for averaging. For estimation of bouton size (capacitance), transient currents from a test pulse were fitted with the sum of two exponentials, with time constants (t), amplitude contributions (A), and corresponding capacitances [$C = (At)/\Delta V$], with the major component representing the charging of the terminal[30]. Test pulses recorded in the cell-attached configuration immediately before break-in were averaged and subtracted from those recorded in the whole-cell configuration (10 consecutive sweeps in each configuration). This important step eliminated any small uncompensated residual capacitance artifacts from the whole-cell trace to allow the most accurate estimation of bouton capacitance[36]. AP threshold was defined as the voltage at which the rate-of-change of the action potential reached 20 mV/ms. For action potential broadening analyses, data were normalized to the value for the first AP of a train for each recording. All values are mean ± SEM.

**Statistics and reproducibility**. For group analyses cells were categorized as 4–6w if they were 28-42 DPI, and 8 + w if >56 DPI. One 27 DPI mossy fiber terminal recording was included in the 4–6w dataset. Cell age-related physiological differences were analyzed by regression and group differences were identified by ANOVA and followed up with Holm-Sidak multiple comparisons tests. If data were non-normal, group differences were identified by a Kruskal-Wallis test with Dunn's post-hoc test. To facilitate comparison with data presented in graphs, most statistical analyses are described in the figure legends. For all analyses, statistical significance was defined as $P < 0.05$. Where differences were statistically significant, effect sizes were estimated by computing Hedges' $g$ values. Sex differences were examined but no consistent differences were observed and therefore the data were pooled for all analyses.

**Reporting summary**. Further information on research design is available in the Nature Portfolio Reporting Summary linked to this article.

## Data availability

The data for all graphs and analyses are provided as Supplementary Data 1.

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

## Acknowledgements

This work was supported by the Natural Sciences and Engineering Research Council of Canada (JSS) and the Michael Smith Foundation for Health Research (JSS).

## Author contributions

N.P.V. performed experiments and analyzed data. J.S.S. provided funding. Both N.P.V. and J.S.S. designed experiments and wrote the paper.

## Competing interests

The authors declare no competing interests.
