## [Peer Review File · Communications Biology]

Reviewers' comments:

Reviewer #1 (Remarks to the Author):

In the revised version of the manuscript, the authors have addressed some of the issues that were raised in the previous round of reviews.

However, several of the major issues are still open.

This short study is still highly descriptive and still provides little insight beyond reporting the shape of action potential waveforms in adult-born MFBs. This major issue was raised by all 3 reviewers (R1: "the contribution of this paper is relatively small"; R3: "it is unlikely to inform current thinking about information processing due to the narrow scope of the current experiments.") Together with R3 (their major point #2), we have previously suggested that the scope of this study could be broadened by assessing the conductance densities that would be required in immature vs mature MFBs to produce the relatively uniform action potential shapes. In the revised version of the manuscript, the authors now "speculate" about Nav and Kv densities in immature vs mature MFBs. Given that all reviewers agree that the scope of this manuscript could and should be broadened, this issue merits to be addressed by more than just speculation.

As already suggested in the previous review, simulations using a simple compartmental model of the MFB & axon that is constrained by the authors' data could go a long way in addressing this issue. The morphology of the MFBs and the action potential dynamics are well constrained by the authors' data. Nav and Kv channel kinetics have been studied in detail in MFBs and have previously been incorporated into MFB models (e.g. PMID 15694327; though the kinetics would need to be scaled for the higher recording temperatures). Even in the absence of additional o-o patch experiments, these constraints should allow to at least establish ranges of Kv and Nav densities in immature vs mature boutons. For example, the upstroke of APs strongly constrains Nav conductance densities when morphology and Nav gating kinetics are known (see e.g. PMID 6329543), as is the case here.

Another major issue that still persists concerns the relatively low amplitudes of APs reported in this study. The authors argue that "any filtering was the same between groups.". That is certainly correct - however, if any differences between groups occur in a frequency range beyond the corner frequency of the series resistance / pipette capacitance filter, then these differences will be missed. For example, assuming a series resistance of 100 MOhm and a pipette capacitance of 7pF, the corner frequency will be approximately 250Hz (by estimating it very roughly as $f_c = 1/(2 \pi RC)$). If any differences occur at frequencies >0.25kHz - which is plausible for signals that are as fast as central action potentials - then they will be missed by the recording system.

Again, this issue could be addressed by simulations, by filtering the simulation output with a (simulated) RC filter corresponding to the typical pipette capacitance and series resistance when fitting the simulation output to the experimental data. Such simulations would reveal if any potential differences in AP waveform might be lost by low-pass filtering. Additionally, the authors may want to test whether AP waveforms between immature and mature MFBs are still uniform if only recordings with the lowest Rs are included in the analysis.

Reviewer #2 (Remarks to the Author):

The authors of this manuscript address the development of excitability and the action potential waveform at mossy fiber terminals of adult-born neurons, finding that the AP waveform exhibits a mature phenotype prior to maturation of passive intrinsic excitability.

This is a revised manuscript. The authors have improved it by addressing all the technical concerns raised in the prior reviews. The main result that young adult-born neurons exhibit a mature AP waveform in terminals is well-supported and novel. I don't have any further concerns beyond the narrow scope of the work that has not changed.

Minor correction/typo in the following sentence:

"Similar to results from somatic recordings (Fig. 1), boutons from young adult-born neurons fired action potentials with less current injection (had enhanced rheobase)."

Less current injection in young neurons = lower rheobase = enhanced excitability.

It would be useful to also report the voltage threshold for APs.

Please see our responses to the reviewer comments below, in bold.

Reviewer #1 (Remarks to the Author):

In the revised version of the manuscript, the authors have addressed some of the issues that were raised in the previous round of reviews.

However, several of the major issues are still open.

This short study is still highly descriptive and still provides little insight beyond reporting the shape of action potential waveforms in adult-born MFBs. This major issue was raised by all 3 reviewers (R1: "the contribution of this paper is relatively small"; R3: "it is unlikely to inform current thinking about information processing due to the narrow scope of the current experiments.") Together with R3 (their major point #2), we have previously suggested that the scope of this study could be broadened by assessing the conductance densities that would be required in immature vs mature MFBs to produce the relatively uniform action potential shapes. In the revised version of the manuscript, the authors now "speculate" about Nav and Kv densities in immature vs mature MFBs. Given that all reviewers agree that the scope of this manuscript could and should be broadened, this issue merits to be addressed by more than just speculation.

As already suggested in the previous review, simulations using a simple compartmental model of the MFB & axon that is constrained by the authors' data could go a long way in addressing this issue. The morphology of the MFBs and the action potential dynamics are well constrained by the authors' data. Nav and Kv channel kinetics have been studied in detail in MFBs and have previously been incorporated into MFB models (e.g. PMID 15694327; though the kinetics would need to be scaled for the higher recording temperatures). Even in the absence of additional o-o patch experiments, these constraints should allow to at least establish ranges of Kv and Nav densities in immature vs mature boutons. For example, the upstroke of APs strongly constrains Nav conductance densities when morphology and Nav gating kinetics are known (see e.g. PMID 6329543), as is the case here.

Every study generates additional questions and we appreciate that the reviewers would like to see more data. There will always be more potential experiments but, with limited resources, we have to prioritize those questions that align with our interests (and also experiments that are needed to validate our findings, see below). Here we had a single goal, which was to assess excitability and AP waveform in the terminal. Additional experiments to determine channel densities is beyond the scope of what we set out to do. We don't discount the importance of these experiments though and so in this revision we have included our underlying data as supplementary data. This way, readers who are interested in performing these simulations can use our data to do so.

Another major issue that still persists concerns the relatively low amplitudes of APs reported in this study. The authors argue that "any filtering was the same between groups.". That is certainly correct - however, if any differences between groups occur in a frequency range beyond the corner frequency of the series resistance / pipette capacitance filter, then these differences will be missed. For example, assuming a series resistance of 100 M Ω and a pipette capacitance of 7pF, the corner frequency will be approximately 250Hz (by estimating it very roughly as $f_c = 1/(2 \pi RC)$). If any differences occur at frequencies $>0.25\text{kHz}$ - which is plausible for signals that are as fast as central action potentials - then they will be missed by the recording system.

Again, this issue could be addressed by simulations, by filtering the simulation output with a

(simulated) RC filter corresponding to the typical pipette capacitance and series resistance when fitting the simulation output to the experimental data. Such simulations would reveal if any potential differences in AP waveform might be lost by low-pass filtering.

We agree this is an important issue that needed to be addressed and thank the reviewer for bringing this up. The Reviewer's estimate calculation of corner frequency reveals the critical importance of eliminating the effects of pipette capacitance—if left uncompensated, the resolution of the recording system is far too low to resolve fast signals such as action potentials, even with lower resistance pipettes. However, in standard patch-clamp experiments and current-clamp recordings, pipette capacitance is electrically cancelled using positive feedback circuits of the amplifier (after taking care to physically reduce capacitance as much as possible—using thick-walled glass, etc). In our current study we have used the capacitance neutralization circuit of the Multiclamp 700B amplifier to cancel 104% of our fast pipette capacitance (our compensation was 7.2 pF with 6.9 pF C_p) (see Ritzau-Jost et al., 2001). Therefore, residual uncompensated capacitance was very small. However, cancellation is imperfect and residual capacitance is probably never zero, and therefore some errors in measurement (through RC filtering) are expected to occur. The important question is, how large are those errors? Recent studies have thoroughly investigated this by comparing recorded signals to known waveforms under various recording conditions (Ritzau-Jost et al., 2021; Olah et al., 2021). Our recording conditions are very similar to those described by Ritzau-Jost et al for their brain slice recordings from boutons. They systematically evaluated the filtering effects of a 7.5 pF pipette capacitance on fast action potentials (0.32 ms half-duration) from a model circuit recorded by various amplifiers. For a Multiclamp 700, capacitance neutralization of 7.6 pF (101%) produced recorded action potentials with ~9% error in half-duration but up to 38% reduction in peak amplitude. These differences may fully explain the size of the action potentials recorded in our study.

Nevertheless, to further validate our results, we repeated key measurements under conditions far less impacted by RC filtering—which is critically dependent on the frequency of the recorded signals, with slower signals being less affected. We therefore recorded action potentials at 22 degrees C (RT), taking advantage of the markedly slower kinetics in comparison to those recorded at near-physiological temperature (half-duration ~1.2 ms at RT in Vyleta and Jonas, 2014). These data are presented in the new Figure S6. We measured large APs for all three groups of mossy fibers (120, 117, 120 mV for 4-6 wk, >8 wk, and neonatal-born, respectively; n = 10 recordings for each group). Our data are therefore consistent with more pronounced RC filtering of the faster action potentials recorded at ~32 degrees C (although it remains possible that biological factors underly some reduction in AP height with increased temperature (Yu et al., 2012)). Importantly, in agreement with our conclusions from the original dataset, we found no statistically significant differences in amplitude, half-duration, or maximal rising slope of action potentials between the groups (quantification in figure legend).

Additionally, the authors may want to test whether AP waveforms between immature and mature MFBs are still uniform if only recordings with the lowest R_s are included in the analysis.

As suggested by the Reviewer, we also compared peak amplitude, half-duration, and maximal rising slope of APs for bouton recordings from the original (32°C) dataset with the lowest series resistance (below the median of 92 MΩ). No statistically significant differences were found between the three groups (Kruskal Wallis $P = 0.22$, 0.42, and 0.25 respectively), nor specifically between young or mature adult-born cells ($P = 0.47$, 0.42, and 0.36 respectively). Taken together, we believe that our data

provide strong support for the conclusion that bouton action potential waveforms are not distinguishable between these groups of neurons.

Reviewer #2 (Remarks to the Author):

The authors of this manuscript address the development of excitability and the action potential waveform at mossy fiber terminals of adult-born neurons, finding that the AP waveform exhibits a mature phenotype prior to maturation of passive intrinsic excitability.

This is a revised manuscript. The authors have improved it by addressing all the technical concerns raised in the prior reviews. The main result that young adult-born neurons exhibit a mature AP waveform in terminals is well-supported and novel. I don't have any further concerns beyond the narrow scope of the work that has not changed.

Minor correction/typo in the following sentence:

"Similar to results from somatic recordings (Fig. 1), boutons from young adult-born neurons fired action potentials with less current injection (had enhanced rheobase)."

Less current injection in young neurons = lower rheobase = enhanced excitability.

Thanks – this has been corrected.

It would be useful to also report the voltage threshold for APs.

This is a good suggestion. We have now analyzed “first AP threshold” for the minimum current experiments shown in Fig. 2G. No differences in voltage threshold were found between groups of boutons (new supplemental Figure S2). We interpret these results to mean that the voltage threshold is the same at the three groups of boutons, but that threshold is reached with less current in terminals from young adult-born cells due to the increased input resistance of the membranes. We have added a statement about this in the Results section.

REVIEWERS' COMMENTS:

Reviewer #1 (Remarks to the Author):

In the revised version of the manuscript, the authors have added some new analyses and have rephrased some paragraphs. I do not have any further concerns beyond the narrow scope of the work that has not changed.